# Reproduction study - DECAF: Generating Fair Synthetic Data Using Causally-Aware Generative Networks

## Reproducibility Summary

**Scope of Reproducibility**

We conduct a reproduction study on the paper *DECAF: Generating Fair Synthetic Data Using Causally-Aware Generative Networks* by van Breugel et al. [2021]. We investigate the following claims: (1) DECAF is versatile and can remove undesired bias using several definitions of fairness including Fairness Through Unawareness (FTU), Demographic Parity (DP) and Conditional Fairness (CF) measured by FTU and DP metrics (2)DECAF is able to generate fair synthetic data with better quality compared to other existing methods as measured by precision, recall and AUROC of a downstream classifier. (3) DECAF relies on the provision of causal structure in the form of a Directed Acyclic Graph (DAG) and DECAF is robust to incorrect DAG specification and any DAG from the Markov equivalence class found through causal discovery can be used for data generation.

**Methodology**

We repeated the main experiments of the authors using two data sets and added some experiments. The provided author's code was mostly used as a base, but we found various discrepancies between the paper and the code. Also, the provided code was not able to generate results yet and we had to alter it to do that. Therefore, our codebase consists of a combination of new and old code.

**Results**

The overall results from our study show that the DECAF method proposed by the authors removes undesired bias as claimed by the authors. We could reproduce the broad trends as reported by the authors for the Adult data set, but not for the Credit data set. Our results clearly show that DECAF could generate fair synthetic data while still maintaining high downstream utility for the first data set.

**What was easy**

The paper is intuitively written and includes clear graphs that supplement the explanation of the method. The paper also includes extensive appendices. The data sets that were used for the experiments were also provided, and easy to find.

**What was difficult**

The main difficulty was working with the code provided by the authors, which was an incomplete implementation of the DECAF pipeline and did not work yet. There were also various discrepancies between the paper and the code, e.g. additional terms in the training loss and the implementation of a XGBoost classifier instead of a Multilayer Perceptron. We put in much effort to debug and complement the code, which required a significant amount of work and took more time than expected.

**Communication with original authors**

The original authors answered some of our questions regarding the insufficient code and confirmed the code base is still under development. They kindly gave us additional code, but unfortunately this did not solve our problems.

# 1 Introduction

Machine learning models reflect the data on which they have been trained. As a result, the models might also reflect undesired biases in the data and potentially perpetuate the bias present in the data[Lu et al., 2020]. Having undesired biases could have adverse consequences, which could even be amplified by a feedback loop that strengthens the biases already present in the data when people act towards the bias in the algorithm prediction. For example, some crime data set may have historically unwanted bias towards a protected attribute ethnicity. Police officers could then determine some neighborhoods to be "problematic neighborhoods" based on the ethnicity of its inhabitants. If they would patrol those neighborhoods at a disproportionate rate based on their status, they would most likely report more crimes there, which will highlight the purported problem status for this ethnicity group even more. So later instances of crime might reinforce the initially biased data in this case.

Fairness in machine learning has been a major focus area of research in recent years because of the challenges it poses and the huge societal impact it has. Fairness in machine learning can be approached broadly from two different perspectives. One can develop fair learning algorithms which can detect bias in the data and create fair predictors. A large body of work including Kilbertus et al. [2017] and Hardt et al. [2016] have adopted this approach. Alternatively, one can innovate effective methods to debias the data or generate fair synthetic data. The second approach is particularly advantageous because any downstream model trained using the fair data can be guaranteed to satisfy desired fairness requirements irrespective of whether the model is inherently fairness-aware or not.

Many methods have been proposed and can be found in literature for fair synthetic data generation. Fairness-aware GAN (FairGAN) proposed by Xu et al. [2018] offers a very flexible framework for fair data generation but it does not allow for post-hoc changes of the synthetic data distribution. Xu et al. [2019a] propose a causal architecture for generating fair data but its application is limited as it can not be easily extended to multiple protected attributes. It is in this context, van Breugel et al. [2021] proposed **DE**biasing **CA**usal **F**airness (DECAF): a GAN-based model for generating fair synthetic data which represents the true data generating process using a Structural Causal Model (SCM).

# 2 Scope of reproducibility

The paper claims that DECAF is compatible with several definitions of fairness and it achieves high downstream utility of the generated data. Its inference-time fairness allows for post-hoc changes of the synthetic data distribution and is quite flexible for evolving definitions of fairness. The authors also provide theoretical guarantees on the generators convergence and the fairness of downstream models.

We identified the following claims as central to the contributions of the paper:

- *Claim 1*: DECAF is versatile and can remove undesired bias using several definitions of fairness including Fairness Through Unawareness (FTU), Demographic Parity (DP) and Conditional Fairness (CF) measured by FTU and DP metrics.

- *Claim 2*: DECAF is able to generate fair synthetic data with better quality compared to other existing methods as measured by precision, recall and AUROC of a downstream classifier.

- *Claim 3*: DECAF relies on the provision of causal structure in the form of a Directed Acyclic Graph (DAG) and DECAF is robust to incorrect DAG specification and any DAG from the Markov equivalence class found through causal discovery can be used for data generation.

Claim 1 is the most important differentiating factor that makes DECAF a class apart from the rest of the methods. The flexibility to choose fairness definitions is at the core of the flexible framework that the paper offers. Claim 2 is quite fundamental to fair synthetic data generation and the adoption of DECAF for real world applications depends on its validity. We plan to run experiments using Adult data set and Credit Approval data set [Dua and Graff, 2017] that the authors have used and additionally Communities and Crime data set [Dua and Graff, 2017].

Claim 3 is more involved and knowing causal structure of the data is a very strong assumption that the paper makes and it is very important to prove that DECAF produces good results from any of the Markov compatible DAGs generated using any causal discovery method. For the experiments, we plan to use TETRAD [Spirtes et al., 2019] for generating the DAGs using the causal discovery algorithm FGES [Ramsey et al., 2017].

## 3  DECAF: The Method

DECAF is a GAN-based fair synthetic data generator and consists of multiple generators and a discriminator. There are two stages: the training stage and generation (inference) stage. In the training stage a causally-informed GAN learns the causal conditionals in the data. The features are generated in order of the underlying DAG, where each feature is modeled by its own generator. Each synthetic sample is passed to the discriminator, which is trained to distinguish real from synthetic samples. The generator and discriminator share a typical minmax objective:

$$max_{\{G_i\}_{i=1}^d} min_D \mathbb{E}[log\, D(G(Z)) + log(1 - D(X))], \tag{1}$$

with generator $G_i : \mathbb{R}^{|Pa(X_i)|+1} \to \mathbb{R}$, discriminator D: $\mathbb{R}^d \to \mathbb{R}$, $Z$ the distributional parameter of a feature excluding the causal parents and $X$ sampled from the original data. $Pa(X_i)$ denotes the causal parents of $X_i$.

At the inference stage, the learned conditionals are used to generate fair data by removing specific edges from the underlying DAG of the data.

## 4  Methodology

### 4.1  Data sets

We reproduced the experiments of the paper on the Adult data set and Credit Approval data set from Dua and Graff [2017]. For both these data sets, the ground-truth DAG is unknown.

The Adult data set contains over 65.000 samples and has 14 attributes. Its prediction task is to determine whether a person makes over $50K every year and there is a known bias between *sex* and *income* in this data set.

The Credit Approval data set concerns credit card applications and contains 690 instances with 15 attributes each. Because there are some rows with missing values in this data set which we removed, we eventually end up with 678 instances. The attributes have been changed to meaningless symbols and there is no known bias between attributes. Therefore, we inject a synthetic bias into the data for this experiment.

In addition, we use the Communities and Crime data set from Dua and Graff [2017], which combines socio-economic data, law enforcement data, and crime data. It contains around 2000 samples and has 128 attributes. The prediction task is to determine whether a given community is violent or not. This is determined by the number of violent crimes per population. Since this data set already contains bias, it is not necessary to inject it ourselves. The ground truth DAG for this data set is also unknown, but can be found using TETRAD [Spirtes et al., 2019]. The choice for this data set as additional data set was made, because it is one of the most widely used data sets in the fairness domain as stated by Mehrabi et al. [2021].

### 4.2  Experimental setup

We tried to keep the experimental setup close to the paper by performing the two experiments as described in the paper: one on the Adult data set (Experiment 1) and one on the Credit Approval data set (Experiment 2).

In Experiment 1, we perform DECAF on the biased Adult data set and treat `sex` as the protected attribute. Synthetic data is generated using the different fairness definitions and a separate MLP is trained on this data to classify the income. The utility of the generated data is measured by how well the downstream classifier performs on the true data distribution which can be evaluated using a hold-out set of the original data set.

In addition to Experiment 1 as performed by the authors, we repeated this experiment using a different DAG, which we discovered by the causal discovery algorithm FGES [Ramsey et al., 2017] using TETRAD [Spirtes et al., 2019]. We repeated this experiment with a different DAG in order to be able to test Claim 3 further and see if the DECAF results still hold up with a different appropriate DAG.

In Experiment 2, bias is directly injected into the Credit Approval data set to show (with the same data set without bias as benchmark) that DECAF can remove (injected) bias. The bias is synthetically injected by decreasing the probability that someone with a specific ethnicity could get their credit approved. Similarly to the paper, we chose everyone with ethnicity class 4 to be the minority ethnicity group. We test this for different fractions of injected bias. The same DAG as the one used in the paper is used.

In Experiment 3, we repeat the approach of Experiment 1 on a different data set to verify the authors claims about DECAF more thoroughly. In this experiment on the Communities and Crime data set, the attribute `racepctblack` was set to protected attribute since there is already bias for this attribute in the data set. This attribute states the percentage of black people in a given community. The underlying DAG was found using TETRAD [Spirtes et al., 2019], which, due to the large number of nodes, resulted in a number of 970 edges.

## 4.3 Model descriptions

For each of these experiments, we have trained multiple relevant models. Firstly, the baseline classifier was trained on the original data itself, such that we are able to report the predictions of the original data and evaluate them against the multiple DECAF models, namely a new model for every definition of fairness. We were able to discriminate between the different definitions of fairness by removing edges in the DAG. So each of these models were trained with the same MLP Classifier, but the difference is the synthetic data these models were trained on, because for the generation of the synthetic data a different dictionary with the biased edges was specified, depending on the fairness definition. Therefore, we got multiple models such as DECAF-ND, DECAF-FTU, DECAF-CF, DECAF-DP, each indicating on which synthetic the classifier is trained. Additionally, for Experiment 1 we obtained various models that function as baseline models, namely GAN [Goodfellow et al., 2014], GAN-PR, WGAN-GP [Arjovsky et al., 2017] and WGAN-GP-PR. The PR stands for protected removal, which entails that the protected attribute is removed.

## 4.4 Code base

The authors of the article have provided a code base containing DECAF and some toy examples that train the model on small, artificial data. We use it basis for our code, but make some substantial changes. Firstly, we complemented the code to include all experiments. Secondly, we made some changes to follow the paper more, because there were multiple discrepancies between the paper and the code. For example, we changed the implementation of DECAF to also generate the labels of synthetic data, because this is needed to impose the fairness criteria. After all, if no labels are generated, then no edges can be removed.

The code base is written in PyTorch-Lightning and Python, and is structured per model, and there is a separate file where the different data sets are read and processed. [1]

## 4.5 Hyperparameters

We set the hyperparameters as described in the paper. We used the Adam optimizer with a learning rate of 0.001 for 50 epochs and update the generator once for every 10 discriminator updates. We hold out a test set of 10% for all experiments and we train a MLP using the default `scikit-learn` hyperparameters. We chose a test set of 10% since the data sets used are relatively small and we wanted to have enough data for training. For the Credit data set we trained for 250 epochs instead of 50, because convergence was not achieved after 50 epochs. All experiments are repeated 10 times with different seeds.

We use the DAGs of both the first two data sets as given in the paper and for Communities and Crime data set it is discovered by the causal discovery algorithm FGES [Ramsey et al., 2017] using TETRAD [Spirtes et al., 2019]. Additionally, for the Adult data set we trained using another DAG we found in the same way with TETRAD [Spirtes et al., 2019].

## 4.6 Computational requirements

All models were trained on Nvidia GeForce 1080Ti GPUs.

# 5 Results

Overall, only some results support the claims from the original paper. We were able to generate new synthetic data using the DECAF pipeline for different data sets, showing that DECAF is compatible with different definitions of fairness by removing edges in the causal graph. However, we were not able to observe better fairness metrics after

---

[1]Codebase can be found at `https://anonymous.4open.science/r/DECAF-615B`

applying DECAF for every data set. Furthermore, we found that our data quality results from different benchmarks (e.g. GAN [Goodfellow et al., 2014] and WGAN [Arjovsky et al., 2017]) are higher than reported by the original authors, which contradicts the claim that DECAF generates fair synthetic data of higher quality.

The results are shown for each separate experiment. They are expressed in 5 metrics that express the data quality and fairness. The metrics for data quality are precision, recall and AUROC. These range from 0 to 1, where 1 indicates a perfect score. The AUROC is the area under the ROC curve of the downstream model (MLP in all experiments) that is trained on the synthetic data and indicates the performance of the downstream model. The fairness metrics are FTU and DP, which indicate perfect fairness at 0. The FTU metric indicates the direct influence of the protected attribute on the prediction and the DP indicates the difference between the predictions of a downstream classifier between the different classes of the protected variable.

## 5.1 Experiment 1

Table 1 and 2 show the data quality and fairness metrics (respectively) after bias removal on the Adult data set of the original experiment and our own. In particular, we see that the FTU and DP scores go down when removing more biased edges, while the data quality scores decrease. So, as expected, the fairness increases at the cost of data quality. This observation is in line with the original authors and supports Claim 1. However, Claim 2 cannot be verified. Our benchmark models (GAN, WGAN-GP, GAN-PR, WGAN-GP-PR) report similar (if not higher) data quality metrics than DECAF, unlike the original authors. These models do report lower scores on fairness than DECAF. Another unexpected trend we notice is that the recall increases when we remove more biased edges.

| Model | Precision | | Recall | | AUROC | |
|---|---|---|---|---|---|---|
| | Original | Our | Original | Our | Original | Our |
| Original data | $0.920 \pm 0.006$ | $0.883 \pm 0.010$ | $0.936 \pm 0.008$ | $0.932 \pm 0.014$ | $0.807 \pm 0.004$ | $0.914 \pm 0.002$ |
| GAN | $0.607 \pm 0.080$ | $0.803 \pm 0.033$ | $0.439 \pm 0.037$ | $0.928 \pm 0.067$ | $0.567 \pm 0.132$ | $0.754 \pm 0.051$ |
| WGAN-GP | $0.683 \pm 0.015$ | $0.843 \pm 0.039$ | $0.914 \pm 0.005$ | $0.889 \pm 0.063$ | $0.798 \pm 0.009$ | $0.818 \pm 0.025$ |
| GAN-PR | $0.632 \pm 0.077$ | $0.835 \pm 0.019$ | $0.509 \pm 0.110$ | $0.839 \pm 0.085$ | $0.612 \pm 0.106$ | $0.756 \pm 0.047$ |
| WGAN-GP-PR | $0.640 \pm 0.019$ | $0.804 \pm 0.022$ | $0.848 \pm 0.028$ | $0.928 \pm 0.043$ | $0.739 \pm 0.034$ | $0.690 \pm 0.017$ |
| DECAF-ND | $0.780 \pm 0.023$ | $0.810 \pm 0.023$ | $0.920 \pm 0.045$ | $0.906 \pm 0.044$ | $0.781 \pm 0.007$ | $0.802 \pm 0.011$ |
| DECAF-FTU | $0.763 \pm 0.023$ | $0.793 \pm 0.024$ | $0.925 \pm 0.040$ | $0.927 \pm 0.039$ | $0.765 \pm 0.010$ | $0.792 \pm 0.010$ |
| DECAF-CF | $0.743 \pm 0.022$ | $0.758 \pm 0.003$ | $0.875 \pm 0.038$ | $0.985 \pm 0.012$ | $0.769 \pm 0.004$ | $0.707 \pm 0.019$ |
| DECAF-DP | $0.781 \pm 0.018$ | $0.754 \pm 0.002$ | $0.754 \pm 0.002$ | $0.987 \pm 0.016$ | $0.672 \pm 0.014$ | $0.641 \pm 0.032$ |

Table 1: The data quality metrics after the bias removal experiment on the Adult data set.

| Model | FTU | | DP | |
|---|---|---|---|---|
| | Original | Our | Original | Our |
| Original data | $0.116 \pm 0.028$ | $0.045 \pm 0.015$ | $0.180 \pm 0.010$ | $0.205 \pm 0.013$ |
| GAN | $0.023 \pm 0.010$ | $0.057 \pm 0.077$ | $0.089 \pm 0.008$ | $0.132 \pm 0.123$ |
| WGAN-GP | $0.120 \pm 0.014$ | $0.056 \pm 0.045$ | $0.189 \pm 0.024$ | $0.187 \pm 0.100$ |
| GAN-PR | $0.0 \pm 0.0$ | $0.0 \pm 0.0$ | $0.120 \pm 0.012$ | $0.161 \pm 0.063$ |
| WGAN-GP-PR | $0.0 \pm 0.0$ | $0.0 \pm 0.0$ | $0.078 \pm 0.014$ | $0.115 \pm 0.042$ |
| DECAF-ND | $0.152 \pm 0.013$ | $0.047 \pm 0.019$ | $0.198 \pm 0.013$ | $0.205 \pm 0.020$ |
| DECAF-FTU | $0.004 \pm 0.004$ | $0.014 \pm 0.009$ | $0.054 \pm 0.005$ | $0.151 \pm 0.014$ |
| DECAF-CF | $0.003 \pm 0.006$ | $0.017 \pm 0.008$ | $0.039 \pm 0.011$ | $0.035 \pm 0.013$ |
| DECAF-DP | $0.001 \pm 0.002$ | $0.009 \pm 0.007$ | $0.001 \pm 0.001$ | $0.013 \pm 0.012$ |

Table 2: The fairness metrics after the bias removal experiment on the Adult data set.

## Using a different DAG

Table 3 shows the results of experiment 1 on a different DAG. The fairness still increases when using different fairness metrics and the data utility numbers are similar to the first experiment. This supports Claim 3.

| Model | Data Quality | | | Fairness | |
|---|---|---|---|---|---|
| | Precision | Recall | AUROC | FTU | DP |
| DECAF-ND | $0.853 \pm 0.016$ | $0.924 \pm 0.020$ | $0.867 \pm 0.015$ | $0.037 \pm 0.021$ | $0.200 \pm 0.022$ |
| DECAF-FTU | $0.852 \pm 0.012$ | $0.931 \pm 0.019$ | $0.858 \pm 0.007$ | $0.010 \pm 0.007$ | $0.169 \pm 0.011$ |
| DECAF-CF | $0.760 \pm 0.006$ | $0.995 \pm 0.006$ | $0.634 \pm 0.035$ | $0.005 \pm 0.005$ | $0.018 \pm 0.009$ |
| DECAF-DP | $0.759 \pm 0.007$ | $0.995 \pm 0.006$ | $0.570 \pm 0.040$ | $0.010 \pm 0.011$ | $0.013 \pm 0.010$ |

Table 3: Experiment 1 using a different DAG

## 5.2 Experiment 2

Our results of performing the second experiment can be found in Figure 1. The data quality metrics are in Figure 1a, 1b, and 1c and the results of the fairness metrics are in Figure 1d and 1e. We see that all models perform fairly similar in terms of the data utility metrics, which is also what the authors of the DECAF paper found. However, they overall had slightly higher scores for these metrics, with precision around 0.94 and recall around 0.91 over all bias strengths. Their AUROC score was around 0.89, but vastly dropped to 0.70 when moving towards a bias strength of 1 for the DECAF-ND case. We also do not see this drop in our results.

As for the fairness metrics, our results significantly differ from those of the authors. The first major difference we see is that the DP score is not lower for the DECAF-FTU and DECAF-DP case than for the DECAF-ND case. Therefore, in our reproduction of the experiment we were not able to remove bias as measured by the DP metric in the same way the authors of the DECAF paper were able to. As can be seen in Figure 1e our DP score is consistently around 0.075, while the DP score of the authors for the DECAF-ND experiment went up to about 0.5 as the bias strength became 1.

This brings us to the second major difference, which is that the scores for both bias metrics for the DECAF-ND experiment stayed very constant over the amount of bias injected, while we would have expected the FTU and DP score to go up for this experiment when more bias is injected, similarly to what the authors saw.

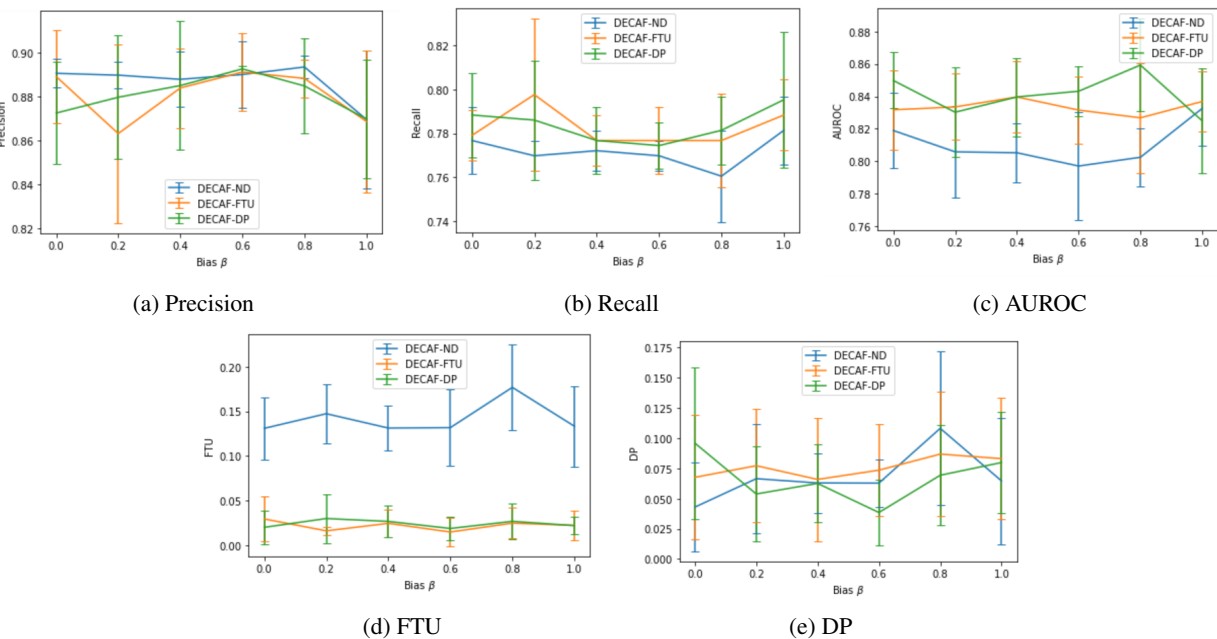

(a) Precision  (b) Recall  (c) AUROC

(d) FTU  (e) DP

Figure 1: Plot of precision, recall, AUROC, FTU, and DP over bias strength $\beta$.

## 5.3 Experiment 3

The results of Experiment 3 are shown in Table 4. When more biased edges are removed, the fairness metrics improve overall, e.g. the FTU of DECAF-ND is 0.482, while the FTU of DECAF-CF is 0.079. In addition, the data quality

metabolics are decreasing when more biased edges are removed, e.g. the precision decreases from 0.649 of DECAF-ND to 0.437 for DECAF-DP. This is also expected from the trade-off between data quality and fairness, as mentioned previously. However, two remarks must be made. Firstly, low data quality is undesirable for a downstream classifier. For illustration, the same classifier will have a precision of 0.901 on the original data set, but 0.437 on the synthetic data generated by DECAF-DP, which is a high cost. Secondly, the FTU metric is higher for DECAF than for the original data, while the DP metric of DECAF is lower. We think further hyperparameter tuning is necessary for making DECAF work for this dataset.

| Model | Data Quality | | | Fairness | |
|---|---|---|---|---|---|
| | Precision | Recall | AUROC | FTU | DP |
| Original data | $0.901 \pm 0.008$ | $0.801 \pm 0.022$ | $0.950 \pm 0.003$ | $0.010 \pm 0.006$ | $0.604 \pm 0.014$ |
| DECAF-ND | $0.649 \pm 0.134$ | $0.672 \pm 0.094$ | $0.765 \pm 0.089$ | $0.482 \pm 0.113$ | $0.551 \pm 0.158$ |
| DECAF-FTU | $0.529 \pm 0.119$ | $0.664 \pm 0.192$ | $0.665 \pm 0.132$ | $0.094 \pm 0.079$ | $0.220 \pm 0.057$ |
| DECAF-CF | $0.462 \pm 0.073$ | $0.695 \pm 0.185$ | $0.594 \pm 0.114$ | $0.079 \pm 0.059$ | $0.091 \pm 0.078$ |
| DECAF-DP | $0.437 \pm 0.085$ | $0.654 \pm 0.180$ | $0.514 \pm 0.124$ | $0.122 \pm 0.059$ | $0.141 \pm 0.102$ |

Table 4: The Data Quality and Fairness metrics after the bias removal experiment on the Communities and Crime data set

# 6 Discussion

From our results as presented above we conclude that the claims listed in Section 2 are partly supported. We will discuss each claim individually:

- *Claim 1*: This claim was partly supported by our results. The results of Experiment 1 supported this claim as well as the results of Experiment 1 with a different DAG. However, our reproduction of Experiment 2 did not support these claims as the DP metric was not lowered for our DECAF-FTU and DECAF-DP model.

- *Claim 2*: We have to look at the results of Experiment 1 for this claim, where we saw our benchmarks report similar or higher data quality metrics than DECAF, but they do perform worse on the fairness metrics. If we only look at these benchmarks we used the claim should be constrained to: DECAF is not able to generate data of better quality, but of better fairness than other existing methods. If we take into account the results for FairGAN [Xu et al., 2018] on the same experiment as presented in the DECAF paper and assume them to be true this claim would be supported as we find higher data utility scores for DECAF than those.

- *Claim 3*: This claim was supported by our results from performing Experiment 1 with a different DAG that we generated using TETRAD [Spirtes et al., 2019] method for causal discovery. The results show that DECAF performs similarly for any DAG from the Markov equivalence class.

## 6.1 Experiments

With regards to Experiment 1, our benchmark models could generate synthetic data of higher quality than those of the authors both for the regular and the PR cases, as can be seen in Table 1. We believe that with proper hyperparameter tuning, a GAN or a WGAN-GP should be able to generate synthetic data with a quality close to the original data and we do not expect any debiasing in the generated data.

Another unexpected thing we saw in Experiment 1 was that the recall went up when performing the experiment with different fairness definitions, while all data utility metrics decreased for the authors. We think this is due to the unbalanced data set with many more positive than negative samples and we think the downstream classifier highly prefers the positive label, causing very little false negatives and thus a high recall but a worse performance for the other metrics.

We were not able to reproduce Experiment 2, we think this might be because of the very small data set size. We made a train-test split with a test size of $10\%$ of the data, which is extremely small for a data set of 678 instances. For future research on the reproducibility of the DECAF paper we would like to try with cross-validation to see if this improves our results of Experiment 2. Moreover, even with the small test set we would not expect the bias metrics to return

constant results over the amount of injected bias for the DECAF-ND experiment. This suggests that the way we injected bias might not be correct.

Experiment 3 partially supports the effects we have seen earlier, where the fairness criteria overall improve, as the fairness definition becomes stricter. However, the FTU metric of the original data performs suspiciously well, meaning that the DECAF model does not provide better fairness metrics than the original data. Furthermore, we again see the data utility versus fairness trade-off happening. However, the data utility metrics are decreasing rapidly as the fairness definition gets stricter. This steep drop suggests that DECAF might not work well on all data sets, or that different hyperparameters are needed for different datasets.

## 6.2 What was easy

The paper is well-written and made it easy for us to understand both the context and the main contributions of the research. More specifically, the preliminaries and required definitions of fairness are clearly explained. This provides an excellent start in the theory. In addition, there are many appendices that provide relevant background and prove additional claims of the paper. For example, the authors explain why protected variable removal is a sub-optimal method for satisfying FTU fairness. Even though the appendices do not contribute to the main claims of the paper, they made it easy to put this research paper into context.

## 6.3 What was difficult

The main difficulty we faced, was getting the code provided by the authors to work. There were two main problems. Firstly, the toy examples did not work. These examples trained DECAF on a small, artificial data set, but the DECAF model was not able to learn and it generated synthetic data with extremely low variance. After debugging the training steps and the model itself, we found that the loss is a large negative value and did not decrease over the epochs. Eventually, we found that the authors made strong assumptions about feature ranges in their implementation and the data has to be preprocessed and normalized to [0,1] for it to work. We managed to solve the problem by using a MinMax scaler, but finding the root cause did take more time than anticipated.

The second problem arose from multiple discrepancies between the paper and the code. There were additional settings mentioned in the paper that were not present in the code, such as splitting the data in a train and test. Additionally, there were extra terms in the code that were not present in the paper, such as additional terms in the training loss.

We could not get the FairGAN code the authors have shared to work.

## 6.4 Future research

As mentioned before, the DECAF model originally provided in the code base of the authors did not learn well and could only generate data with low variance. When looking into this issue, we found existing research describing similar problems when training GANs on tabular data (consisting of discrete and continuous data) [Xu et al., 2019b]. The article proposes mode-specific normalization, which (simply put) entails that each attribute in the data is normalized independently. This model (called CTGAN) supposedly deals better with imbalanced discrete columns and ensures stable learning on tabular data. We made an attempt to include the described mode-specific normalization into DECAF, but did not succeed due to time constraints. It would be interesting future research to include a CTGAN in DECAF, because it might improve the data quality of the synthetic data when DECAF learns better.

## 6.5 Communication with original authors

We contacted the original authors with some questions regarding the code base they have provided online. They replied shortly after and confirmed the code base is still under development. Even though they could not directly answer each question, they kindly shared additional code to help us further. However, since this code was not intended to publish (yet), it was not clearly structured and it was unclear which files were directly related to the paper. Nonetheless, it gave us a better understanding of the perspective and work of the authors, and the Python package (PyCausal) was brought to our attention this way, which is able to obtain graphs of data sets.

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
