# OpenReview forum: "Reproduction study - DECAF: Generating Fair Synthetic Data Using Causally-Aware Generative Networks"
_ML_Reproducibility_Challenge/2021/Fall — Reject_

### Official Review · Reviewer_arg3 · 2022-03-07
**Good Reproduction Attempt**

**Rating:** 7
**Confidence:** 3

**Review:**

Summary: The authors do a nice job of a thorough attempt to replicate the published results of the paper. The inability of the authors to replicate the result appears to reflect a general lack of replicability in the main paper. While no ablation study or extension was provided, I think this is reasonable given the difficulty in producing the main results.

Code: Authors attempted to use the authors' code and hyperparameter search.

Ablation Study: None, though as I say above, I think that is reasonable in this case.

Discussion on results: I would have liked to see some additional discussion around the results/lack thereof. The authors do a nice job detailing the difficulty in reproduction, but it would be nice if additional intuition was provided for this.

Recommendations for reproducibility: the authors do a nice job of detailing their process (which I found reasonable), which I think at least implicitly constitute the basis of a set of recommendations.

Results beyond the paper : None, though as I mention above, I think this is reasonable in this case.

Overall organization and clarity: No clear grammatical issues. Good writing and organization.

---

### Official Review · Reviewer_n29Z · 2022-03-29
**Review of Reproduction study - DECAF: Generating Fair Synthetic Data Using Causally-Aware Generative Networks**

**Rating:** 5
**Confidence:** 3

**Review:**

Authors worked on replicating the paper by Boris et al. (2021) on generating fair synthetic data using casually aware generative models. Efforts were put in to replicate the Fairness Through Unawareness (FTU), Demographic Parity (DP) metrics. To verify the claim of generating fair synthetic data made in the original paper, the authors tried to replicate the precision, recall, and AUROC metrics on the downstream classifier.

The results reported are higher in some cases than in the original paper without any explanation for this boost. The authors could not verify the claim of generating fair synthetic data in experiment 1. Similarly, while replicating the results in the experiment 2 authors could not get a similar drop in the AUROC score as seen in the original paper.

The authors communicated with the original authors to clarify the codebases, and the original authors helped them by providing additional codebases. The unexpected trends observed by the authors as reported in their experiments can further be cleared by communicating with the original authors.

Overall, while the experiments and the paper are well written, and the shared codebase would be helpful to future researchers, the work could not reproduce the original paper completely. The authors can improve reproducibility in the future with thorough communication with the original authors.

Also, I list some typos in the following:
- Line 6: (2)DECAF → missing space
- Line 93: 65.000 → 65,000
- Line 124: set to protected → set to a protected
- Line 140: use it basis → use it as a basis

---

### Meta-Review · Program_Chairs · 2022-04-07

**Recommendation:** Reject
**Confidence:** 5

**Metareview:**

While the report itself is well-written, the fact that the authors were unable to reproduce the original paper makes it difficult to accept.  Through further work and communication with the authors of the original paper, this can possibly be fixed.

---

### Decision · Program_Chairs · 2022-04-09

Reject